# ALCAM: A Novel Surface Marker on EpCAM^low^ Circulating Tumor Cells

**DOI:** 10.3390/biomedicines10081983

**Published:** 2022-08-16

**Authors:** Rossana Signorelli, Teresa Maidana Giret, Oliver Umland, Marco Hadisurya, Shweta Lavania, John Lalith Charles Richard, Ashley Middleton, Melinda Minucci Boone, Ayse Burcu Ergonul, Weiguo Andy Tao, Haleh Amirian, Anton Iliuk, Aliya Khan, Robert Diaz, Daniel Bilbao Cortes, Monica Garcia-Buitrago, Harrys Kishore Charles Jacob

**Affiliations:** 1Department of Surgery, Miller School of Medicine, University of Miami, Miami, FL 33136, USA; 2Sylvester Comprehensive Cancer Center, Miller School of Medicine, University of Miami, Miami, FL 33136, USA; 3Department of Biology, Miller School of Medicine, University of Miami, Miami, FL 33136, USA; 4Department of Radiation Oncology, Miller School of Medicine, University of Miami, Miami, FL 33136, USA; 5Diabetes Research Institute, Miller School of Medicine, University of Miami, Miami, FL 33136, USA; 6Department of Biochemistry, Purdue University, West Lafayette, IN 47907, USA; 7Institute of Engineering in Medicine, University of California San Diego, San Diego, CA 92093, USA; 8Biospecimen Shared Resource, University of Miami, Miami, FL 33136, USA; 9Tymora Analytical Operations, Innovations, West Lafayette, IN 47906, USA; 10Department of Pathology and Laboratory Medicine, Miller School of Medicine, University of Miami, Miami, FL 33136, USA; 11Sylvester Comprehensive Cancer Center, Cancer Modeling Shared Resource, Miller School of Medicine, University of Miami, Miami, FL 33136, USA

**Keywords:** biomarkers, pancreatic cancer, CTC, surfaceome, proteomics, flow cytometry, ALCAM

## Abstract

**Background:** Current strategies in circulating tumor cell (CTC) isolation in pancreatic cancer heavily rely on the EpCAM and cytokeratin cell status. EpCAM is generally not considered a good marker given its transitory change during Epithelial to Mesenchymal Transition (EMT) or reverse EMT. There is a need to identify other surface markers to capture the complete repertoire of PDAC CTCs. The primary objective of the study is to characterize alternate surface biomarkers to EpCAM on CTCs that express low or negligible levels of surface EpCAM in pancreatic cancer patients. **Methods:** Flow cytometry and surface mass spectrometry were used to identify proteins expressed on the surface of PDAC CTCs in culture. CTCs were grown under conditions of attachment and in co-culture with naïve neutrophils. Putative biomarkers were then validated in GEMMs and patient samples. **Results:** Surface proteomic profiling of CTCs identified several novel protein biomarkers. ALCAM was identified as a novel robust marker in GEMM models and in patient samples. **Conclusions:** We identified several novel surface biomarkers on CTCs expressed under differing conditions of culture. ALCAM was validated and identified as a novel alternate surface marker on EpCAM^low^ CTCs.

## 1. Introduction

Pancreatic cancer has the highest mortality rate among all cancers. For all stages combined, the 5-year survival rate is 11% [1]. Distant organ metastasis is responsible for nearly 90% of cancer-associated deaths [2], with the liver and surrounding peritoneum being major sites for metastasis. It has been reported that circulating tumor cells (CTCs), which are disseminated neoplastic cells shed from the primary tumor, have clonal capacity to reach distant organs and initiate tumor growth [3,4,5]. Therefore, it is valuable to analyze CTCs to further understand the phenomenon of metastasis. More specifically, in pancreatic ductal adenocarcinoma (PDAC), tumor cells that are shed from the primary site enter the bloodstream, either as singlets or clusters, and promote tumor survival even after surgical resection [4]. CTCs also play an important role in the diagnosis and detection of pancreatic cancer [6,7,8,9]; enumeration [10,11,12,13,14,15] and characterization of CTCs have great potential for clinical prediction, diagnosis, and monitoring in patients with PDAC [16]. CTCs can be conveniently harvested from patients via minimally invasive phlebotomy; however, their rarity among billions of blood cells makes their identification and isolation challenging [7,8]. While recent studies have proposed several advances in instrumentation and strategies to detect CTCs in the blood stream, there has been a heavy dependence on a limited toolbox of markers to characterize these CTCs. The Epithelial Cellular Adhesion Molecule (EpCAM) has been the singular most widely used marker for separating CTCs; current methods [17,18,19,20,21,22,23,24,25,26,27] employ a positive enrichment strategy to isolate these cells. They are limited to a unique subset of cells in a heterogeneous population of CTCs, either as singlets or as clusters that have high levels of surface EpCAM expression. Methods that do not employ a positive enrichment approach [28,29,30,31,32,33,34,35,36] and are solely based on the physical characteristics of the cells have several issues such as low processing volumes and times in addition to requiring tedious sample fractionation before separation [8]. These different strategies have captured efficiencies ranging anywhere from 11% to 92% [2,8], which still makes it quite a herculean task to capture the complete panorama of CTCs that are released from each cancer type, with a 32.5% detection rate in pancreatic cancer [37]. A retrospective solid tumor study comparing CTC detection rates with EpCAM reports a 32.5% detection rate for PDAC CTCs [37]. Currently, the most widely used method for capturing CTCs from PDAC patients has relied heavily on immunocapture techniques that require positive enriching for cells that express cytokeratin and EpCAM and negative enriching for CD45 [38,39,40,41,42,43]. The lack of appropriate methods to cultivate CTCs in vitro poses a major hindering block in characterizing surface or other markers in CTCs.

Therefore, there is an immediate need to identify other surface biomarkers to be used in tandem with EpCAM to capture the complete repertoire of cells in circulation. The present study used flow cytometry and mass spectrometry to identify surface proteins on CTCs that express low or negligible levels of EpCAM. CTCs were cultured under different conditions of attachment and also in co-culture with neutrophils to identify transitory changes on the surface proteome. ALCAM that was identified by both methods was validated in patient samples and showed great promise of being an additional biomarker that can be used for screening CTCs in patients with metastatic pancreatic cancer.

## 2. Materials and Methods

### 2.1. Patient-Derived Tumor Cell lines

Multiple CTC lines were obtained from Celprogen (Torrance, CA, USA). These cells were labeled with PE-conjugated Anti EpCAM (BioLegend Cat#324206, San Diego, CA, USA) and analyzed by flow cytometry to determine the level of surface EpCAM. The three selected CTC cell lines (CM61, CF49, and HM59) did not have any surface level expression of EpCAM. Cell lines were obtained from the following patients: Caucasian Male aged 61 (CM61), Caucasian Female aged 49 (CF49), and Hispanic Male aged 59 (HM59). CTC cells were isolated from the peripheral blood of PDAC patients with liver metastases (mets). An equal number of cells for each patient were cultured in Medium 106 (Thermo Cat# M106500, Waltham, MA, USA) with hydrocortisone (1 μg/mL), EGF (10 ng/mL), FGF (10 ng/mL), heparin (10 μg/mL), and 2% FBS and then shifted to a medium with 2% exosome-depleted FBS before harvesting for EVs. Cells were cultured under adherent and non-adherent conditions using normal or ultra-low attachment plates (Corning Cat# CLS3814-24EA, Corning, NY, USA), respectively, to mimic homogenous populations of CTCs in circulation. Cells were also cultured with naïve neutrophils to mimic interactions with blood cells.

### 2.2. Naïve Neutrophil Isolation

Naïve neutrophils were isolated from human blood that was collected in EDTA tubes; 50 mL of whole blood was used for each biological replicate. The MACSexpress whole blood neutrophil isolation kit (Miltenyi Cat#130-104-434, Bergisch Gladbach, Germany) was used to isolate neutrophils. The purity of neutrophils was confirmed by Wright Giemsa staining and flow cytometry. The purity was close to 99% on flow cytometric characterization of CD11b^+^ cell populations.

### 2.3. Flow Cytometric Surface Protein Characterization

The LEGENDScreen Human PE kit (BioLegend Cat#700007) was used to characterize the cell surface markers expressed on the CTC cell lines. To facilitate the analysis of all three cell lines, CM61 was unlabeled, CF49 was labeled with CellTrace violet (violet, 405 nm excitation, filter 450/45), and HM59 with CellTrace far red (red, 638 nm excitation, filter 660/20). PE-conjugated antibodies (yellow, 561 nm excitation, filter 585/42) present in each well assessed the expression values of the cell surface markers. The cells were prepared according to the manufacturer’s instructions. The samples were acquired on a Beckman Coulter CytoFlex S equipped with 4 laser lines, 405, 488, 561, and 638 nm, using a 96-well plate reader. CytExpert 2.3 (Beckman Coulter, Brea, CA, USA) was used for the acquisition of data.

### 2.4. Karyotyping

Conventional methods were followed for the identification of G-banding pattern in the CTCs. The cells were karyotyped at the Cytogenetics and Molecular Diagnostic laboratory and the Pathology Services at the University of Miami, Mailman Center for Child development.

### 2.5. Preparation of Surface Proteome

CM61 cells were cultured either as adherent cultures or on low attachment plates (Corning Cat#CLS3814), following which they were either scraped or pelleted to isolate cells for surface proteome preps. Then, 4 × 10^7^ cells were taken for each prep. The CM61 CTC cell line was grown in low attachment plates. The cell line was subsequently mixed with naïve neutrophils in a 1:1 ratio in HBSS medium (Thermo Cat#14185052) for 1 h; the cells were then pelleted and subjected to surface proteome preps. The Pierce Cell Surface Protein Isolation kit (Thermo Cat#89881) was used for all cell proteome labeling and isolation. The media were removed and cells were washed twice with ice-cold PBS. Surface proteins were cross linked with Sulpho-NHS-SS-Biotin in ice-cold PBS for 30 min at 4 °C. The reaction was quenched and washed with TBS twice. Cell lysis was performed by lysis buffer provided in the kit and cells were sonicated on ice using five one-second bursts. Cells were then incubated on ice for 30 min and vortexed every 5 min for 5 sections. Samples were then centrifuged at 10,000 g for 2 min at 4 °C. The supernatant was incubated with NeutrAvidin agarose slurry and incubated for 60 min with end-over-end mixing using a rotator. The beads were then washed twice, after which they were eluted with a buffer containing 8 M urea and 50 mM DTT.

### 2.6. Preparation of Samples for LC-MS

The immunoprecipitated samples were processed by Tymora Analytical Operations (West Lafayette, IN, USA). The proteins were incubated at 37 °C for 15 min to reduce the cysteine residues and alkylated by incubation in 100 mM iodoacetamide for 45 min at room temperature and in the dark. The samples were diluted 3-fold with 50 mM triethylammonium bicarbonate and digested with Lys-C (Wako Cat # 121-05063) at 1:100 (wt/wt) enzyme-to-protein ratio for 3 h at 37 °C. The samples were further diluted 3-fold with 50 mM triethylammonium bicarbonate, and trypsin was added to a final 1:50 (wt/wt) enzyme-to-protein ratio for overnight digestion at 37 °C. After digestion, the samples were acidified with trifluoroacetic acid (TFA) to a pH < 3 and desalted using Top-Tip C18 tips (Glygen Cat#NC9980572, Washington, DC, USA), according to the manufacturer’s instructions. A portion of each sample was used to determine peptide concentration using Pierce Quantitative Colorimetric Peptide Assay (Thermo Fisher, Waltham, MA, USA). The samples were completely dried in a vacuum centrifuge and stored at −80 °C. Based on the concentration, 1.4% of each peptide sample was analyzed by Liquid Chromatography Mass Spectrometry (LC-MS).

### 2.7. Transmission Electron Microscopy

CM61 CTCs were grown in either low attachment or adherent conditions. Adherent cells were grown on cover-slips while suspension cells were pelleted and fixed. They were both fixed overnight in 2% glutaraldehyde in 0.1 M phosphate buffer, post-fixed for 1 h in 2% osmium tetroxide in 0.1 M phosphate buffer, dehydrated through a series of graded ethanols, and embedded in EM-bed (Electron Microscopy Sciences, Fort Washington, PA, USA). The glass cover-slip was dissolved in hydrofluoric acid. Then, 100 nm sections were cut on a Leica Ultracut EM UC7 ultramicrotome (Wetzlar, Germany) and stained with uranyl acetate and lead citrate. The grids were viewed at 80 kV in a JEOL JEM-1400 transmission electron microscope (Tokyo, Japan) and images were captured by an AMT BioSprint 12 digital camera (Woburn, MA, USA).

### 2.8. LC-MS/MS Analysis

Dried peptide and phosphopeptide samples were dissolved in 4.8 μL of 0.25% formic acid with 3% (vol/vol) acetonitrile; 4 μL of each was injected into an EasynLC 1000 (Thermo Fisher Scientific, Waltham, MA, USA). Peptides were separated on a 45 cm in-house packed column (360 μm OD × 75 μm ID) containing C18 resin (2.2 μm, 100 Å; Michrom Bioresources, Auburn, CA, USA). The mobile phase buffer consisted of 0.1% formic acid in ultrapure water (buffer A) with an eluting buffer of 0.1% formic acid in 80% (vol/vol) acetonitrile (buffer B), run with a linear 60- or 90-min gradient of 6–30% buffer B at a flow rate of 250 nL/min. The Easy-nLC 1000 was coupled online with a hybrid high-resolution LTQ-Orbitrap Velos Pro mass spectrometer (Thermo Fisher Scientific). The mass spectrometer was operated in the data-dependent mode, in which a full-scan MS (from *m*/*z* 300 to 1500 with the resolution of 30,000 at *m*/*z* 400) was conducted, followed by MS/MS of the 10 most intense ions (normalized collision energy—30%; automatic gain control (AGC)—3 × 10^4^, maximum injection time—100 ms; 90 s exclusion). To specifically focus on the proteins present only on the surface and to remove all background proteins with cytoplasmic or nuclear localization, the biotin-enriched proteome was compared with a previously published in silico and cell surface protein database [44]. Only bona fide cell surface proteins were considered for further studies.

### 2.9. Proteome Discoverer Label-Free Quantitation Analysis

The human Swiss-Prot database with no redundant entries was used to search the raw files, using Byonic (Protein Metrics, Cupertino, CA, USA) and Sequest search engines in Proteome Discoverer 2.3 software (Thermo Fisher Scientific). The MS1 and MS2 mass tolerances were set at 10 ppm and 20 ppm, respectively. In the processing workflow, search criteria for both search engines were performed with the following criteria: full trypsin/P digestion, a maximum of two missed cleavages for peptides allowed, a static modification of carbamidomethylation on cysteines (+57.0214 Da), and variable modifications of oxidation (+15.9949 Da) on methionine residues and acetylation (+42.011 Da) at N terminus of proteins. The false-discovery rates (FDR) cutoffs of proteins and peptides were set at 0.01. For the quantification, the intensities of peptides were extracted with minimum peak count as 1, PSM confidence FDR of 0.01 as strict and 0.05 as relaxed, and maximum RT shift as 5 min. For calculations of protein abundance, the sum of sample abundances of the connected peptide groups was added together and used for downstream analysis. All protein and peptide identifications were grouped, and any redundant entries were removed. Unique peptides and unique master proteins were reported. Statistical analyses were performed in Perseus software version 1.6.5.0 (Max Planck Institute, Planegg, Germany) [45,46]. The normalized intensities of proteins were extracted from Proteome Discoverer search results and log-based 2 transformed. The abundances were categorized into their respective categories. The proteins with a maximum of 30% missing values were kept. The imputation for the missing abundances was performed by assigning small random values from the normal distribution with a downshift of 1.8 SDs and a width of 0.3 SDs. All abundances for each protein were further normalized by subtracting each protein or phosphoprotein abundance with the most frequent value. Then, the ANOVA test was performed, and the permutation-based FDR was calculated for each protein.

### 2.10. GEMM Animal Experiments

The study was conducted with approval from our Institutional Animal Care and Use Committee (IACUC protocol number 21-045). Kras^LSL^.^G12D/+^; p53^R172H/+^; Pdx1Cre^tg/+^ (KPC) mice were generated in a mixed background (SvJae/C57Bl6/BalbC). Tumors were allowed to be initiated and to develop. Animals were sacrificed at a humane endpoint and pancreas and distant organ metastases were collected. Samples were fixed in 10% neutral buffered formalin and processed by conventional methods.

### 2.11. CTC and DTC Isolation from KPC Mice

Blood and ascitic fluid were collected from at least 10 KPC mice pooled and run separately on a Parsortix CTC isolation system (AnglePLC, Guildford, UK). CTCs were collected after washing and by reverse flush from the cassette. CTC or DTC were then suspended in Matrigel and injected subcutaneously in C57Bl6 mice. Tumors were allowed to develop for a month, and, after the animals were euthanized, tumors were collected and processed by conventional methods.

### 2.12. HEK 293 and CTC Cell Line Staining

HEK293, CF49, HM59, and CM61 cells were cultured under adherent conditions, trypsinized, and neutralized with media containing FBS. The cells were then captured on filters (faCTChecker (Circulogix, Inc., Hallandale Beach, FL, USA)) [47]. A pre-mild formaldehyde-based fixation step was performed before filtration. After filtration, the cells attached to the filter were washed in 1× PBS (Life Technologies Corporation, Carlsbad, CA, USA). Isolated cells were stained using primary mouse anti-human CD45, leucocyte common antigen (Agilent Technologies, Santa Clara, CA, USA, catalog no. IR75161-2, clones 2B11 + PD7/26) and rabbit anti-CD166 (1:100, Abcam, Cambridge, UK, catalog no. ab109215, clone EPR2759) antibodies; samples were stained overnight at 4 °C. After being washed thrice in PBS for 5 min each, samples were incubated for 1 h at room temperature with Alexa Fluor^®^ 680 conjugated goat anti-mouse (1:500, Invitrogen, Waltham, MA, USA, catalog no. A-21058) and AlexaFluor^®^ 594 conjugated goat anti-rabbit (1:500, Invitrogen, catalog no. A-11037) secondary antibodies. Subsequently, cells were incubated for 1 h at room temperature with primary Pan-Keratin (C11) mouse monoclonal antibody Alexa Fluor^®^ 488 conjugate (1:150, Cell Signaling Technology, Denvers, MA, USA, catalog no. 4523) and primary pan-cytokeratin mouse monoclonal antibody (AE1/AE3) Alexa Fluor 488 conjugate (1:500, eBioscience™, San Diego, CA, USA, catalog no. 53-9003-82). After another washing step, slides were cover-slipped using ProLong™ Gold Antifade Mountant containing 4′,6-diamidino-2-phenylindole (DAPI) (Life Technologies Corporation, catalog no. P36931) for nuclei staining and imaged on the Olympus VS-120 instrument (Olympus, Tokyo, Japan) using a 10× objective.

### 2.13. CTC Immunostaining and Enumeration

In total, 10 mL of peripheral blood samples was processed for the enumeration of CTCs, as described before [48]. Briefly, an automated round-pore microfilter-based isolation system faCTChecker (Circulogix, Inc., Hallandale Beach, FL, USA), developed specifically to capture CTCs, was used for the isolation of CTCs in this study [47]. A pre-mild formaldehyde-based fixation step was performed before filtration. After filtration, the cells attached to the filter were washed in 1× PBS (Life Technologies Corporation, CA). Isolated cells were stained using primary mouse anti-human CD45, leucocyte common antigen (Agilent Technologies, catalog no. IR75161-2, clones 2B11 + PD7/26), and rabbit anti-CD166 (1:100, Abcam, catalog no. ab109215, clone EPR2759) antibodies; samples were stained overnight at 4 °C. After being washed thrice in PBS for 5 min each, samples were incubated for 1 h at room temperature with Alexa Fluor^®^ 680 conjugated goat anti-mouse (1:500, Invitrogen, catalog no. A-21058) and AlexaFluor^®^ 594 conjugated goat anti-rabbit (1:500, Invitrogen, catalog no. A-11037) secondary antibodies. Subsequently, cells were incubated for 1 h at room temperature with primary Pan-Keratin (C11) mouse monoclonal antibody Alexa Fluor^®^ 488 conjugate (1:150, Cell Signaling Technology, Danvers, MA, USA, catalog no. 4523) and primary pan-cytokeratin mouse monoclonal antibody (AE1/AE3) Alexa Fluor 488 conjugate (1:500, eBioscience™, catalog no. 53-9003-82). After another washing step, slides were cover-slipped using ProLong™ Gold Antifade Mountant containing 4′,6-diamidino-2-phenylindole (DAPI) (Life Technologies Corporation, catalog no. P36931) for nuclei staining and imaged on the Olympus VS-120 instrument (Olympus) using a 10× objective. CTCs were then identified using our immunocytochemistry (ICC) criteria (round/ovoid size ≥ 6-mm) as anti-CD166/ALCAM positive/DAPI positive/CD45 negative and cytokeratin positive/DAPI positive/CD45 negative cells with additional morphology criteria.

### 2.14. Enrichment Analyses

Reactome [29] and Ingenuity Pathway Analysis [49] were used for enrichment analysis. For Reactome analysis, a *p* value cutoff of 0.01 or lower was taken to determine the statistically significant enrichment of proteins in any pathway being analyzed. For IPA analysis, multiple datasets were considered for analysis. The first is the list of candidate biomarkers identified in the flow cytometric screen (markers with expression percentages of at least 30% in all three CTC lines combined), followed by an individual list of proteins identified from respective culture conditions from the mass spectrometric analyses. A core analysis was conducted by uploading the dataset of proteins and their abundances. Both direct and indirect relationships were selected for the reference Ingenuity Knowledge Base. All interaction networks were selected and only experimentally verified evidence was selected. Canonical pathways output was selected for identifying the top pathways that could be regulated. The IPA pathway analysis was performed with default settings, with all data sources, all confidence levels, all species, all tissues and cell lines, all mutations, all relationship types, all publication date ranges, all node types, all diseases, and all biofluids selected. ALCAM was selected as the first molecule set for which activity was known. In addition, the following nodes for which activity was unknown were uploaded, which included: WNT/βCatenin, EGF, PI3K/AKT, FAK, ERK/MAPK, and tumor microenvironment pathways. These following nodes were selected due to their relevance and significance in pancreatic cancer and having been reported earlier to be important pathways that are differentially regulated [50,51,52,53,54,55,56,57,58,59]. Murakami, 2019 #12GO analysis of identified proteins was conducted using ShinyGO [60], and an FDR cutoff of 0.05 was applied and the top 20 pathways were reported. The pathway analysis was performed in the Ingenuity Pathway Analysis (IPA) for ALCAM. First, the path explorer feature was used to find both direct and indirect paths between the biomarkers and the pathways. Second, the molecule activity predictor (MAP) feature was used to predict the upstream or downstream effects of the upregulated biomarkers.

### 2.15. Generation of a Mouse–Human Hybrid TMA

A mouse- and human-specific TMA was generated for the rapid validation of biomarkers that needed to be assayed. A majority of the samples were taken from the KPC model of pancreatic cancer, which is one of the most widely used models for disease modeling [61]. Cores were taken from KPC mice pancreas at various stages of development. Samples were taken from both diseased pancreas and adjacent non-neoplastic areas as well as from normal pancreas. Distant organ metastases to lungs, liver, or spleen were also cored onto the TMA. Representative sample cores were also taken from subcutaneous tumor models, their associated metastases, and from subcutaneous models of circulating or dissociated tumor cells. Diseased human pancreas along with normal samples were also taken for assessment in the clinic. Control cores of heart and colon tissue were taken for orientation purposes and to assess staining in other organs. Cores were also taken from an inducible Kras model that was generated by subcutaneously implanting the cells in normal mice. Cores of 1 mm diameter were taken and arrayed into a TMA with multiple duplicate cores in different locations. The detailed TMA map of all sections arrayed on the hybrid mouse–human TMA is shown in Appendix A.

### 2.16. Histochemical Validation of Selected Markers

The following antibodies were used for histochemical validation: anti-ALCAM (Abcam # ab 109215; 1:1000 dilution). The TMA cores were heated at 60 °C for two hours and then conventionally hydrated, following which antigen retrieval was performed by steaming the slides for 20 min in antigen unmasking solution Tris Based (VectorLabs, Mowry Ave Newark, CA, USA, Cat#H3301-250), following which slides were cooled to room temperature. Endogenous peroxidase, pseudoperoxidase, and alkaline phosphatase in FFPE sections were blocked with Bloxall (VectorsLabs Cat # SP6000-100) for 10 min. Cells were then washed in IHC wash buffer (PBS with 0.1% Tween20) for 5 min, following which they were incubated with normal goat serum (2.5%) for 20 min for blocking non-specific sites. The antibody was then diluted in goat serum and the sections were incubated at 4 °C overnight in a humidified chamber. The slides were washed in wash buffer for 5 min and incubated for 30 min with ImmPRESS Universal Polymer Reagent (VectorLabs Cat# MP-7451) for 30 min. Then, the slides were washed twice in wash buffer and incubated with ImmPACT DAB EqV peroxidase substrate solution (VectorLabs Cat#SK4103-400) for 5 min. The slides were then washed in wash buffer twice for 5 min each and then rinsed in tap water. The slides were counterstained with Hematoxylin QS counterstain (VectorLabs Cat# H3404-100) for 60 s and rinsed in tap water. The slides were conventionally dehydrated and then mounted with Vectamount permanent mounting medium (VectorLabs Cat# H5000-60). The slides were checked and scored by an expert pathologist (G-B.M.).

## 3. Results

### 3.1. CTCs Express Multiple Surface Proteins That Might Play a Role in Pancreatic Cancer

Several PDAC CTC cell lines isolated from patients with metastatic cancer were first probed for the expression of surface EpCAM by labeling cells with PE-conjugated anti- EpCAM antibodies. We identified three CTC lines that were negative for EpCAM but were cytokeratin-positive. Attentive care was taken to allow for representation across the Hispanic and Caucasian populations: cell lines were taken from Caucasian female and male patients aged 49 and 61 years, respectively, as well as from a Hispanic male patient aged 59 years. This assisted in providing an unbiased analysis of surface protein expression across different sample populations. A pipeline of the experimental design of the study is shown in Figure 1i. These CTC cell lines had a characteristic of differentiating into two subtypes of cells in culture that could not be separated; single cell sorting would eventually give rise to the other cell type. Immunofluorescent labeling of the cells indicated they were CK+/CD45- (Figure 1ii).

These EpCAM negative cells were subjected to further experiments to characterize the surface proteome. A high throughput cytometric screen was carried out to characterize the surface markers present on all three isolated cell lines. The BioLegend screen contains antibodies against nearly 361 surface proteins alongside respective mouse, rat, or hamster Ig isotype controls. A quick assessment of the cell lines and the repertoire of markers they express [62] was analyzed by flow cytometry. For the simultaneous processing of all three cell lines, the individual cell lines were labeled with one of the CellTrace dyes in either the far red, violet spectrum, or left unlabeled. At least 30,000 events were recorded for each marker analyzed in every cell line used in the study. An arbitrary cutoff of being expressed at least more than 50% in one of the cell lines was followed to identify a candidate list of the top 10 markers expressed across all three cell lines (Table 1). A complete list of all surface markers and their identification percentages across all three CTC lines analyzed are provided in Appendix A. Flow cytometric panels for expressed markers along with their EpCAM status are shown in Figure 1iii. It is interesting to note that less than 10% of the cells were EpCAM-positive and that sorting EpCAM-negative cell lines eventually gave rise to a minor EpCAM-positive population; this could possibly be associated with the dual cell type populations in the cell lines. We, therefore, characterized the cell line as EpCAM^low^ even though more than 90% of the cells were EpCAM-negative. It would be a good starting point to analyze the surface profile of such cells to identify other markers that are expressed at higher levels and can be used in screening cells.

IPA analysis of the top 20 surface markers (combined expression of at least 30% in all three CTC lines) confirmed that these proteins were all plasma membrane-associated proteins. Most of the proteins are transmembrane receptors. Among the other identified types were NT5E, a phosphatase, EPHA2, a kinase, LAMP2, a GPCR, and TNFSF4, an enzyme. A detailed IPA (Qiagen, Hilden, Germany) analysis is shown in Appendix A. It is interesting to note that most of the proteins enriched in this analysis are involved in the activation of mononuclear leukocytes, blood cells, and lymphocytes, or have roles to play in the adhesion of/to immune cells and other cell types. These CTC lines show the characteristics of leukocyte-like cells, which is consistent with the earlier literature on CTCs displaying a hybrid dual positive phenotype [63]. The enrichment of proteins in canonical pathways mediates the cross-talk between dendritic cells, natural killer cells, and the activation of the JAK1, JAK3, STAT3, and TH2 pathways. A detailed list of enriched canonical pathways is shown in Appendix A.

A literature review was also conducted to assess the role of the identified proteins in pancreatic cancer specifically. In pancreatic cancer, several of the identified proteins have a major role to play in progression, immune evasion, and resistance to therapy. CD166/ALCAM is a cell adhesion molecule that mediates both heterotypic cell–cell contacts and has been previously shown to promote T cell activation and proliferation. ALCAM-positive cells have been shown to be highly tumorigenic in comparison to ALCAM-negative cells [64]. The overexpression of ALCAM is also an independent prognosis marker for poor survival and early tumor relapse in PDAC [65], with secreted ALCAM being proposed to be a novel diagnostic marker for pancreatic cancer [66]. MICA and MICB are cell surface proteins; however, unlike canonical class I molecules of the major histocompatibility complex, they do not associate with Beta 2 microglobulin. Levels of MICA/B are also elevated in pancreatic cancer [67] and the inhibition of MICA/B, mediated by constitutive activation of the AMPK-GATA2 axis, may serve as a therapeutic target of pancreatic cancer immune evasion [68]. CD9 is another integral membrane protein associated with integrins that regulate processes such as platelet activation and aggregation, cell adhesion, and paranodal junction formality. It has also been shown to be involved in regulating tumor metastasis in several cancers. In PDAC, CD9-positive cells have tumor-initiating capacity and give rise to tumor heterogeneity [69] with high levels of expression being strongly associated with poor prognosis [70]. EPHA1 is a receptor tyrosine kinase that promiscuously binds to membrane-bound ephrin A family ligands, leading to contact-dependent bidirectional signaling into neighboring cells. In pancreatic cancer, serum and exosomal levels of EPHA1 family member have been identified to be a potential diagnostic marker for PDAC, complementing CA19-9 and CA242 [71,72]. Moreover, 5′-nucleotidase (NT5E or CD73) is an enzyme that hydrolyzes extracellular nucleotides into membrane permeable nucleosides. It is a glycosylphosphatidylinositol (GPI)-anchored protein that attenuates tumor immunity via cooperation with CD39 to generate immunosuppressive adenosine. Therefore, CD73 may function as a promoter of cancer progression and a regulator in immune patterns [73,74,75].

### 3.2. Variation in Biomarkers on Surface Could Be Attributed to the Genetic Makeup of the CTCs

In order to analyze whether the changes in expression of the CTCs were due to genetic amplifications or gene rearrangements, a chromosome analysis was performed on 20 G-banded metaphase cells from multiple unstimulated CTC cultures. A G-banding technique was used to analyze all cells. The CF49 cells were analyzed at a banding resolution of 350–400. A total of 20 metaphase cells were analyzed. The majority of the cells showed hypodiploidy. Three cells had a chromosome number in the range of 54–71, a hyperdiploidy to a near-triploidy. All chromosomes were structurally abnormal, and the details of the structural changes could not be identified nor described (cytogenetic result: 37–46[13]/51–76[7]). HM59 cells were analyzed at a banding resolution of 400–450. A total of 20 metaphase cells were analyzed. The vast majority of the cells in the sample showed hypodiploidy. Three cells had a chromosome number in the range of 69–71, a triploidy or hyper-triploidy. All chromosomes were structurally abnormal, and the details of the structural changes could not be identified nor described (cytogenetic result: 37~44[17]/69~77[3]) [76]. The CM61 cells were analyzed at a banding resolution of 350–400. There were two cell populations identified in this sample. One population with chromosome numbers in the range of 37–45 showed a so-called hypodiploidy in all but one of the cells analyzed. The other population had chromosome numbers in the range of 51–76, resulting in a hyperdiploidy to a near-triploidy. All chromosomes were structurally abnormal, and details of the structural changes could neither be identified nor described (cytogenetic result: 37–46[13]/51–76[7]) [77]. Detailed images of the G-banding are shown in Appendix A. The abnormal chromosomal arrangement in the cells could also be an important factor to consider when analyzing CTCs. The CTC lines have a high degree of genetic variation among themselves, making it difficult to compare all three CTC lines in a head-to-head comparison. We decided to focus on only one of the cell lines, CM61, as a cell line of choice for all downstream proteomic experiments. Differences in the expression levels of markers could very well be attributed to gene rearrangements and amplifications.

### 3.3. CTCs Express Differing Levels of Biomarkers on the Surface When Grown in Different Culture Conditions

CTCs in circulation are seen either as single cells or clustered with other cell types. Additionally, the LEGENDScreen kit is primarily used to screen cell lines and primary cells (such as PBMCs, bone marrow-derived cells, and cells isolated from tissues) for a fixed number of cell surface markers [76,77,78]. This kit restricts which markers can be analyzed and, therefore, limits the potential to identify other surface markers that are more biologically significant. An unbiased mass spectrometric approach was investigated in order to overcome these shortcomings. Since the three cell lines are quite similar in the expression of cell surface markers, the CM61 cell line was used for a detailed characterization of the surface proteome under differing culture conditions. Cells were cultured in low attachment plates as these conditions mimic those seen in in vivo circulation [79,80]. Attentive care was taken to avoid breaking up clumps of cells in order to preserve all surface interactions of the proteins. This was performed to mimic homogenous cluster-like conditions in culture. Cells were also cultured in adherent conditions to represent surface markers of cell lines grown on plastic. CTCs are also identified as clusters; to mimic these conditions, a co-culture with naïve neutrophils was also included in the study. For the sake of experimental ease, the CM61 CTC line was grown in low attachment conditions and mixed with an equal number of neutrophils. In all cases, the experiment was performed in triplicates in order to show stronger statistically significant data. Surface biotinylation of proteins exposed on the surface was carried out by labeling cells with EZ-Link Sulfo-NHS-SS-Biotin, which is a thiol-cleavable amine-reactive biotinylation reagent. Cells were subsequently lysed and enriched using NeutrAvidin agarose beads. In order to identify a bona fide signal from only the CTCs and not from the neutrophil contribution, the biotinylated surface proteome of naïve neutrophils was also analyzed. As the individual proteome of each cell type in co-culture was not labeled, attentive care was taken to select only markers that showed changes in expression levels, which did not reflect the actual background neutrophil surface marker levels. To achieve this, only a log2FC of more or less than 1.5-fold was considered as actual change attributed to the CTC surface proteome in culture. If a molecule did not satisfy the above-mentioned criteria, it was not considered for further analysis. A complete list of molecules identified in all the triplicate mass spectrometric runs is provided in Appendix A.

#### 3.3.1. CTC Surface Markers Differ in Cells Grown in Adherent or Low Attachment Conditions

Transmission Electron Microscopy images of CM61 CTCs that were cultured either under adherent or low attachment conditions were obtained (Appendix A).

No major changes were observed with the cells grown in adherent and low attachment conditions except for membranes being more extended and convoluted while cells grown in suspension had a more rounded smooth surface. We wanted to investigate the surface proteome of cells grown in both conditions to investigate changes in the surface presentation of markers. A co-culture condition with neutrophils was also included to mimic surface changes in proteins when they come in contact with blood cells. Only cells that were cultured in low attachment conditions were used for the co-culture to be close to actual conditions in the blood.

When cells grown in low attachment and adherent conditions were compared, we identified a total of 2046 proteins by mass spectrometric analyses in both conditions, with 77 of them being unique to adherent conditions and 52 being unique to low attachment conditions (Figure 2).

A log2FC of 1.5-fold was used as an arbitrary cutoff to identify molecules that were expressed either higher or lower in each of the conditions. When comparing the low attachment culture conditions to the adherent culture conditions, we identified 93 proteins that were upregulated in low attachment conditions and 147 proteins that were downregulated in low culture conditions (Figure 2). Appendix A provides a list of proteins that were identified in both the adherent and low attachment culture conditions. A Reactome analysis of proteins that were upregulated in the low attachment conditions indicated neutrophil degranulation, nucleotide savage, and laminin interaction pathways as the top three pathways identified. In addition to this, we also identified proteins that regulate ECM proteoglycans, and RHO GTPase, RAC2, and NOTCH pathways. The Reactome analysis was performed with proteins that were downregulated in the low attachment culture condition; the neutrophil degranulation pathway was identified as the most significantly enriched pathway. Additionally, RAB regulation of trafficking, RHOJ and RHO signaling, MET receptor recycling, and processing of SMDT1 pathway proteins were also identified. A list of all Reactome enriched proteins is provided in Appendix A.

#### 3.3.2. CTC Surface Markers Are Differentially Regulated on Exposure to Naïve Neutrophils

CTCs are normally observed as single cells, clusters of cells forming a homogenous population, or heterogeneous clusters with other cells such as neutrophils, platelets, and/or fibroblasts. This cluster association is necessary to evade immune detection and seeding at distant metastatic sites. Earlier studies have looked into transcriptomic changes in CTCs and neutrophils with a cell–cell junction and cytokine receptor pairs being identified through transcriptomics [81]. However, not much is known about cell surface receptors on CTCs and how they are modulated when they come in contact with neutrophils. In order to investigate that aspect, CM61, one of the selected CTC cell lines, was cultured in the presence of naïve neutrophils, and its surface proteome was subsequently examined. Control CTCs were cultured under ultra-low attachment conditions. All experiments were performed to mimic cells in circulation. Furthermore, the surfaceome of naïve neutrophils was taken as a control condition. Only CTCs cultured in low attachment conditions were analyzed for the expression of surface markers.

When the surface proteomes of CTCs grown in low attachment conditions and in co-culture with neutrophils were compared, a total of 2050 proteins were identified in both conditions combined with 80 proteins identified that were unique to neutrophil co-culture and 163 identified that were unique to low attachment conditions. However, upon eliminating proteins that did not show a 1.5-fold difference in overexpression of the neutrophil surface proteome, only 163 upregulated proteins and 619 downregulated proteins that were in co-culture conditions remained. A Reactome analysis of the upregulated proteins enriched for platelet and neutrophil granulation pathways was performed. Major signaling pathways that the proteins enriched for were MAPK oncogenic signaling, MAP2K and MAPK pathways, as well as BRAF and CRAF signaling pathways. Upon CTCs–neutrophils interaction, various pathways were enriched in the downregulated proteins. Some of the top enriched pathways were the neutrophil degranulation pathway, RHO GTPase pathways, Glucokinase regulation, mitochondrial protein import, VxPx cargo targeting to cilium, protein localization, CDH1 auto-degradation, insulin receptor recycling, and protein folding pathways. A complete list of statistically significant pathways is shown in Appendix A.

### 3.4. GO Analysis Indicates a Unique Repertoire of Enrichments in Different CTC Culture Conditions

GO analysis of proteins in the low attachment CTC culture condition indicated that the upregulated proteins were enriched for peptidyl-methionine modifications, negative regulation of cell migration involved in sprouting angiogenesis, positive regulation of TGFβ production, negative regulation of blood vessel endothelial cell proliferation involved in sprouting angiogenesis, and regulation of pattern recognition receptor signaling biological processes to be active. Among the enriched cellular components, the proteins were enriched for secretory granule membrane, Ficolin-1 rich granule and its lumen, secretory granules, and the tertiary granule membrane. Most proteins were involved in increased aminopeptidase activity, peptidase and endopeptidase activation, IgG binding, SH3 domain binding, and modified amino acid binding (Figure 3).

However, the top biological processes enriched with the downregulated proteins included chylomicron remodeling, hydrogen peroxide metabolic processing, neutrophil activation involved in immune response, protein targeting to mitochondria, and regulation of reactive oxygen species metabolic processing. Cellular enriched components were the chylomicron complex, tertiary, specific, and secretory granule proteins. Very interestingly, the proteins involved with lipase inhibitor activity, phosphatidylcholine-sterol O-acyltransferase activator activity, cholesterol binding, and enzyme regulation were downregulated (Figure 3).

When the same analysis was performed with upregulated proteins in co-culture conditions with naïve neutrophils, the resulting enriched biological processes were proteins involved in NADH regeneration and metabolism as well as SRP-dependent co-translational proteins targeting the membrane and ER. These proteins belonged to the Ficolin-1 rich granule lumen, cytosolic ribosome, and melanosome cellular components. This could also be indicative of contamination by ribosomal proteins in the immunoprecipitation with beads, which is a very common problem associated with such methods. Proton-transportation of ATP synthase, proton channel activity, and vinculin cadherin and actin filament binding molecular functions were highly enriched. Among the downregulated genes, several biological processes that regulate myeloid cell activation or degranulation were enriched in addition to processes that control exocytosis, secretion, and protein localization. The cellular components that were enriched were the spliceosome complex or multiple vesicle lumens (Figure 3). A complete list of statistically significant GO annotations is shown in Appendix A.

### 3.5. ALCAM Is a Reliable Marker Expressed in PDAC CTCs

Comparative analysis of protein markers that were identified and detected reliably among all three cell lines (CF49, CM61, and HM5) as well as in the surface proteome of CM61 cells cultured under differing culture conditions was performed. All proteins identified from the mass spectrometric studies in all three conditions were looked into and compared with proteins identified from the flow cytometry screens. In total, we identified 2290 proteins by MS and 354 by flow cytometry (Figure 4i). ALCAM (CD166) was identified as a common marker that appeared in both experimental conditions. ALCAM is a cell adhesion molecule that is known to mediate both heterotypic cell–cell contacts via interaction with CD6 or by establishing homotypic cell–cell contacts. It is a member of the immunoglobulin receptor subfamily with five immunoglobulin-like domains. It has a long extracellular domain (528 amino acids) with a transmembrane region as well as a short cytoplasmic domain (33 amino acids). There are two known isoforms, “short” and “long”, differing in only a stretch of 12 negatively charged amino acids located in the extracellular domain. The two variants are produced by alternate splicing and are differentially susceptible to cleavage by ADAM metallopeptidase domain 17 or 10 (ADAM17/10) [82,83]. It was expressed in nearly 100% of the cell population among almost all of the three CTC lines that were analyzed (Figure 4i). On investigation of the expression levels in the different culture and co-culture conditions of CM61 CTC line, there were several peptides that were identified from ALCAM. Representative quantitation MS1 spectra of all peptides detected for ALCAM are shown in Figure 4ii, while MS2 fragmentation spectra of four ALCAM peptides are shown in Figure 4iii. The ALCAM peptides were detected in higher levels in cells that were cultured either in adherent or non-adherent conditions but did not show much variability. A biological assessment of expression would be in the co-culture conditions where the levels of the peptides show a more biological observable trend; however, these are higher in abundance compared to neutrophil levels of ALCAM, making it a suitable marker for assessment in PDAC CTCs. ALCAM is a potential biomarker that is minimally expressed on neutrophils but overrepresented in CTCs cultured in adherent, low attachment, and co-culture conditions.

### 3.6. Alcam Is Expressed in PDAC GEMM Tumors, Distant Metastatic Sites, CTCs, and DTCs

In order to assess the presence of Alcam in mouse models of cancer, the KPC (Kras^LSL^.^G12D/+^; p53^R172H/+^; Pdx1Cre^tg/+^) mouse model was utilized as it is an established mouse model of pancreatic cancer. A hybrid tissue microarray (TMA) was generated with tissue samples from mouse and human primary tumors and also from distant metastatic sites such as the liver. The TMA also had representative sections of subcutaneous cultures of circulating and disseminated tumor cells (CTCs and DTCs). The Parsortix system was used to separate CTCs from the blood. In-cassette staining did not detect any cells and no cells were able to be grown in culture. To circumvent this issue, the isolated CTCs and DTCs were injected subcutaneously in C57Bl6 mice and allowed to form tumors. In the normal pancreas, Alcam is primarily localized to the cell membrane of the duct, acini, and islet cells. In primary and metastatic PDAC, the localization is shifted to the cytoplasm. The used antibody targeted the cytoplasmic domain of Alcam that recognized both the translated isoforms of Alcam, thus providing an unbiased detection of both isoforms (Figure 5i). Staining with an antibody that targets the extracellular domain shows similar results (data not shown). Pancreas from KPC GEMM mice was also analyzed at different time points of growth. The expression of Alcam was seen as early as 25 days, with the expression becoming more cytoplasmic in localization in older mice aged 3–7 months (Figure 5ii). Interestingly, the CTC and DTC cultures showed membrane and cytoplasmic expression of Alcam; there was very strong staining observed in both the CTC and DTC subcutaneous tumors, which is indicative of Alcam being a very robust marker (Figure 5iii).

### 3.7. ALCAM Is Expressed in Diseased Pancreas and Related Metastases and on the Surface of CTCs Isolated from PDAC Patients

A hybrid TMA was used to analyze the expression of ALCAM in human pancreas and in mets. Representative cores were stained with the antibody recognizing the cytoplasmic terminal of the protein. ALCAM is expressed in the primary tumor and also in liver metastases. Membrane and cytoplasmic staining were observed in primary and liver metastases. This is quite similar to the expression in mice pancreas and metastases (Figure 6i). To clinically investigate the validity of ALCAM, identified from the screens as a potential marker, patient samples were screened and collected that were pathologically verified not to express EpCAM. Peripheral blood was collected from these patients. Blood was collected from a 65-year-old male who did not have any chemotherapy and did not have liver mets, a 57-year-old male with PDAC and extensive liver mets, and a 65-year-old female patient with recurrent PDAC. The blood was then parsed through the Parsortix instrument and CTCs were collected. The CTCs were then fixed in formalin and filtered on Circulogix filters and stained for cytokeratin, ALCAM, and CD45 with DAPI as a nuclear stain. We identified a total of 75 single CTCs and one cluster from the 57-year-old male that were CD45-negative, cytokeratin-positive, and ALCAM-positive. On the other hand, in the 65-year-old male patient, we identified 31 single CTCs and two clusters. In the third patient, with recurrent PDAC, we identified 22 single CTCs and one cluster. Representative images of single CTCs and clusters are shown in Figure 6ii. It is important to note that all the CTCs were positive for ALCAM, which is once again indicative of it being a robust marker for the detection of PDAC CTC.

### 3.8. ALCAM Correlates with Multiple Signaling Pathways in Pancreatic Cancer

In order to investigate the role of ALCAM in pancreatic cancer signaling, an IPA Path Designer (Qiagen) analysis was performed. Common pathways that have a role in pancreatic cancer were taken into consideration: the EGF, FAK, WNT/β-catenin, PI3K/AKT, ERK/MAPK, and the tumor microenvironment pathways. A literature review indicates a direct association of ALCAM with increased activity and expression of AKT [84], RELA [85], and CREB [86]. Indirect associations are noted between the NFκB, FAK, PI3K/AKT, WNT-β-catenin, BCL2, and tumor microenvironment pathways. Hypothetical activation of ALCAM results in the activation of all the signaling pathways associated with cancer. There is, however, the inhibition of MMP2 that could result in the activation of the FAK signaling pathway or the tumor microenvironment pathway. Nonetheless, this last connection is inconclusive due to the lack of valid data points to determine its role in the molecule activity prediction algorithm (Figure 7). Enrichment details for pathways are provided in Appendix A.

## 4. Discussion

Pancreatic cancer has defied early detection and can only be detected in advanced stages. It is curable if detected in stage I, provided it is amenable to resection. There is a need for pancreatic cancer CTC-specific markers to help increase early detection rates and identify the gamut of cellular subtypes in circulation. In the past, EpCAM and cytokeratin levels have been widely used to identify CTCs. However, EpCAM is generally not a robust marker as there are different levels of expression based on the EMT status of the cell. Additionally, as shown in this study and others discussed in the literature, exposure to neutrophils or other cell types changes the expression of cell surface markers. To identify alternate markers on EpCAM negative or low expressing cells, CTCs were screened for the expression of EpCAM. Three cell lines were identified in which less than 10% of the cells expressed surface EpCAM. All CTCs analyzed in this study were identified from patients who had metastatic lesions in the liver and were racially diverse in order to reduce bias on cell surface epitope projection. Since the cells were genetically very diverse, for this study, we decided to focus on only one of the cell lines to exclude the inherent variability bias that could be introduced by sampling cell lines with a variable expression of cell surface proteins. We plan to address this variable genetic makeup and cell surface protein expression in future studies. Additionally, cell lines from both sexes were considered to finally generate a biomarker surface map with minimal bias. To be able to obtain enough cells for flow cytometry or enough proteins for mass spectrometric analyses, only CTCs that could be cultured and expanded were selected for this study. These CTCs were positive for CA19-9, CEA, Galactosyl transferase II, FAD, Alpa-1-antitrypsin, mucin, and CK7, and 3–5% of the cells were also CD133-positive. Epitope presentation is bound to change in circulation when cells associate with different blood cell types. The flow cytometric screen for surface markers identified a few makers, but the original intent of the screen was to quickly characterize the collected blood and cancer cells. It was only through a surface proteomic characterization that we could obtain an unbiased analysis of the CTC cell surface. When CTCs were cultured in adherent and low attachment conditions, most proteins did not change, even though there were differences in some surface proteins. The most drastic difference was observed when a CTC cell line was cultured with naïve neutrophils. For simplicity, only cells cultured under low attachment conditions were used in the co-culture experiments. An earlier study from our group showed that extracellular vesicles (EVs) secreted from CTCs could induce early granulatory changes in neutrophils [87]. We observed the downregulation of nearly 619 proteins; these changes could have been initiated due to cell–cell communication between the neutrophils and the CTCs or due to the transfer of EVs from one cell type to another. This is important when it comes to detecting clusters that evade immune detection as it could explain the difference between CTCs associating with other blood cells or even with other cell types to form clusters. Additionally, these could also be responsible for seeding at distant metastatic sites based on what surface receptors/proteins are present. The circulatory microenvironment with blood cells, immune cells, serum, exosomes from different tissues, growth factors, and other blood constituents complicates the detection of proteins for characterizing patient CTCs. Co-culturing CTCs with other immune or blood cell types could provide additional clues as to what surface proteins would be exposed in heterogeneous clusters. A cytogenetic analysis was also performed to understand the genetic makeup of these cells. There are extensive chromosomal rearrangements and triploidy observed in these culturable CTCs, which is indicative of an increase in marker expression as related to the cellular makeup. Exome sequencing (hybridization-based) or solid tumor panel (amplicon-based) assays could not provide any suitable results. The generated amplicons did not align well with the target, and the majority of the amplicons were on Chromosome 21 (about 95%). No variant or coverage information could be obtained. Therefore, additional experiments need to be performed in order to investigate the genetic makeup of these cells.

ALCAM was identified in both screens and posed as a suitable marker for investigation. However, the role of ALCAM in pancreatic cancer is debated. ALCAM has been detected at the mRNA level in PDAC CTCs in patients after palliative chemotherapy [88]. Nevertheless, we did detect ALCAM in the patient who had recurrent PDAC after their chemotherapeutic regimen. Additionally, its behavior could be explained by possible differences in the transcription and translation of ALCAM, as our earlier study only focused on the transcriptional signature and not the protein level. Moreover, there are two isoforms of the protein that are present; we could overcome the issue by using an antibody that detects the cytoplasmic domain that is identical between the two isoforms. ALCAM has been identified as a good serum [89] and prognostic marker [65], with ALCAM-positive cells being highly tumorigenic and invasive [64,89,90,91,92,93]. Increased expression of ALCAM in PDAC is an independent prognostic marker for poor survival and early tumor relapse [65,94]. Pathway prediction analysis prognosticates that ALCAM has a major role to play in the activation of all major pancreatic cancer-related signaling pathways, making it an important protein to be investigated in CTC panels.

We provide information on novel surface markers on CTCs that express low levels of EpCAM. This would assist in identifying novel classes of CTCs that are negative or express very low levels of EpCAM. In no way does this study eliminate the need for EpCAM as a marker but rather suggests the use of ALCAM and other surface markers in tandem to better aid in detecting all subtypes of cells in circulation. Further validation of ALCAM is currently being carried out on a larger cohort of patients to ascertain its suitability as a routine CTC marker.

## 5. Conclusions

For the first time, we provide a compendium of cell surface markers on PDAC CTCs expressing low or negligible levels of EpCAM, which were cultured in transitory conditions. ALCAM has been identified as a promising PDAC CTC marker in mice and in a smaller repertoire of clinical samples and holds great potential to be used for screening in combination with current epitopes in preclinical and clinical settings pending extensive testing in a larger cohort of PDAC patients.

## Figures and Tables

**Figure 1 biomedicines-10-01983-f001:**
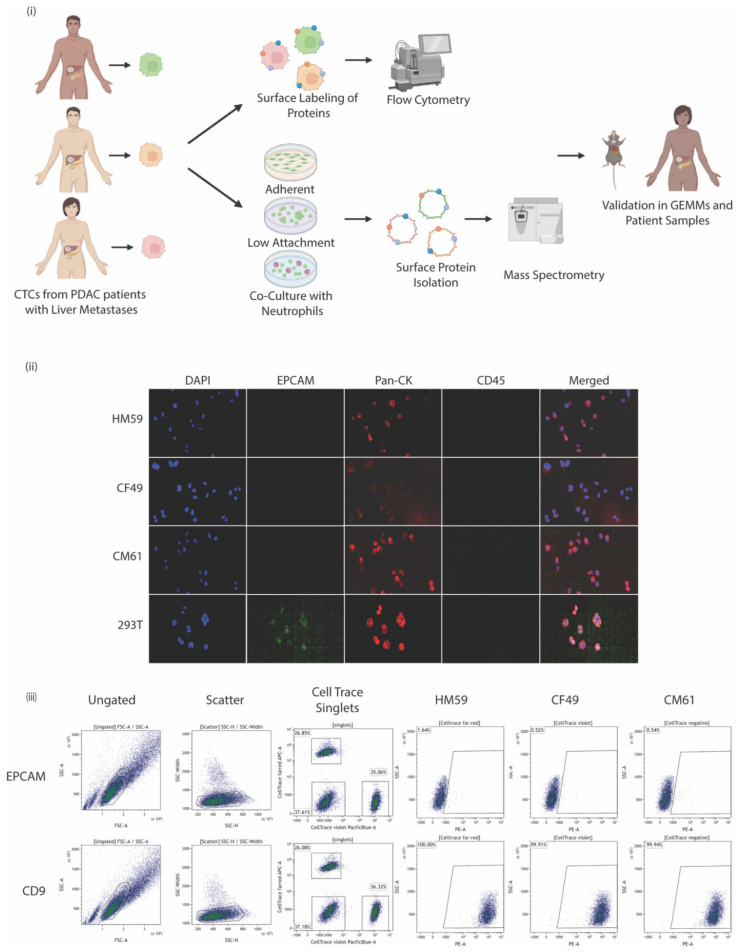
(**i**) Experimental pipeline indicating CTCs were isolated from peripheral blood of patients. The cells were either surface labeled to identify proteins by flow cytometry or were cultured under different conditions: as adherent cells or on low attachment plates as single cells or homogenous clusters, and as heterogeneous clusters in a co-culture with naïve neutrophils. The surface protein was isolated and subjected to mass spectrometry. (**ii**) Immunofluorescence labeling of patient-isolated CTCs compared with control CTCs stained for DAP1 nuclear stain, EpCAM, pan-cytokeratin, CD-45, and a merged composite image of all stains combined. (**iii**) Representative scatter plots for EpCAM, and CD9 staining on CTCs that were subjected to flow cytometry. Ungated, scatter, CellTrace plots and individual plots for the CTCs are shown.

**Figure 2 biomedicines-10-01983-f002:**
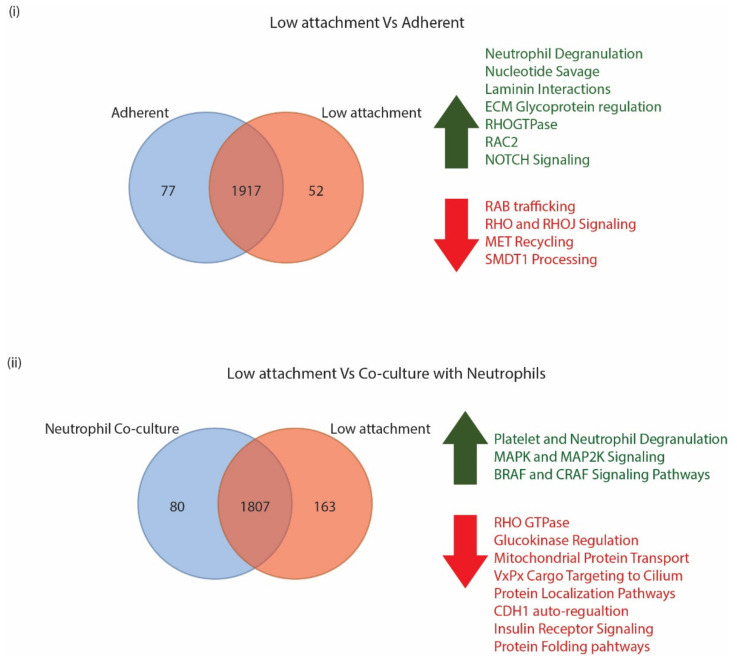
(**i**) Venn distribution of proteins identified in CM61 CTC cultured in adherent and non-adherent conditions along with cellular processes that are upregulated in green and downregulated in red. (**ii**) Venn distribution of proteins identified in CM61 CTC cultured in low attachment and in co-culture with naïve neutrophils along with cellular processes that are upregulated in green and downregulated in red.

**Figure 3 biomedicines-10-01983-f003:**
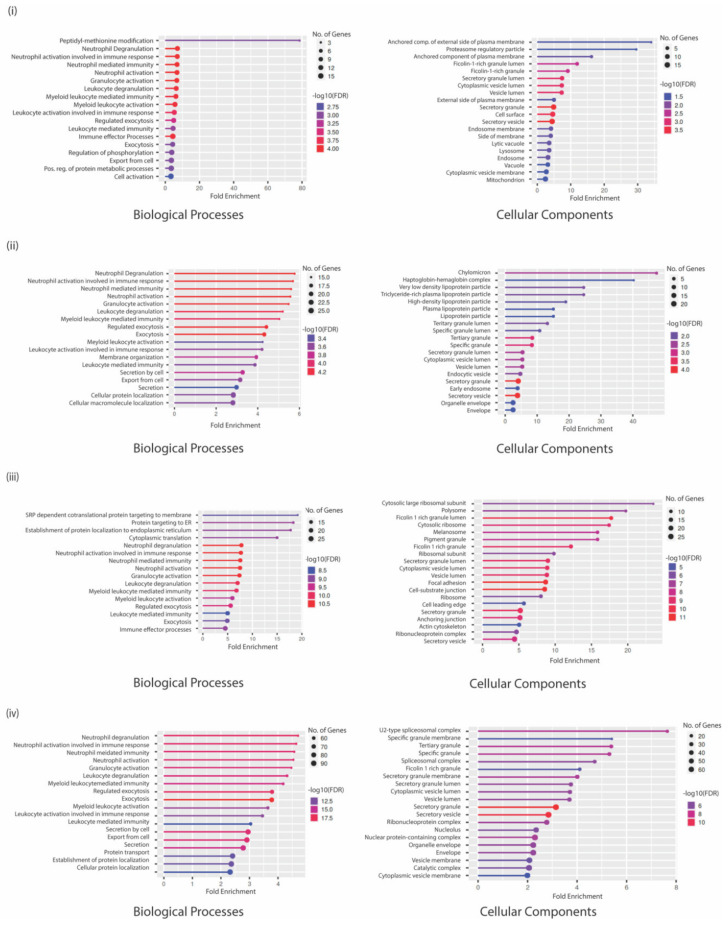
Gene-ontology analysis representing biological processes and cellular components in gene sets that are (**i**) upregulated in CTC low attachment culture conditions; (**ii**) downregulated in CTC low attachment culture conditions; (**iii**) upregulated in co-culture conditions with naïve neutrophils; and (**iv**) downregulated in co-culture conditions with naïve neutrophils. The thickness of the circles represents the number of genes in the cluster while the color of lines represents the FDR enrichment.

**Figure 4 biomedicines-10-01983-f004:**
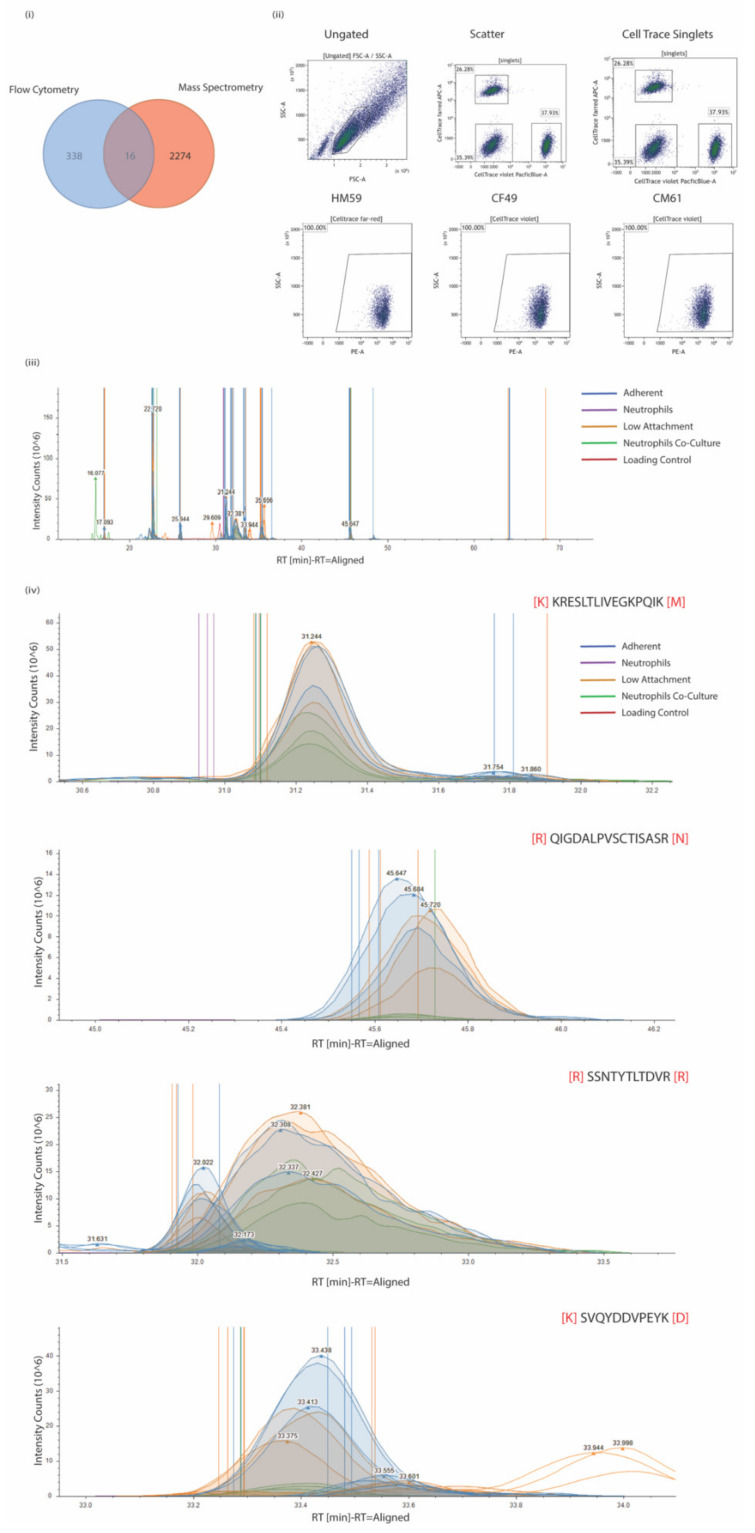
(**i**) Venn distribution of proteins identified from flow cytometry and mass spectrometric studies with an overlap of 16 proteins, of which one is ALCAM (CD166). (**ii**) Representative scatter plots for ALCAM staining on three CTC lines that were subjected to flow cytometry. Ungated, scatter, and CellTrace plots and individual plots for the CTCs are shown. (**iii**) Quantitation MS1 spectra for all peptides identified from ALCAM in mass spectrometric analysis. (**iv**) Fragmentation MS2 spectra of four ALCAM peptides and intensities across different experimental conditions.

**Figure 5 biomedicines-10-01983-f005:**
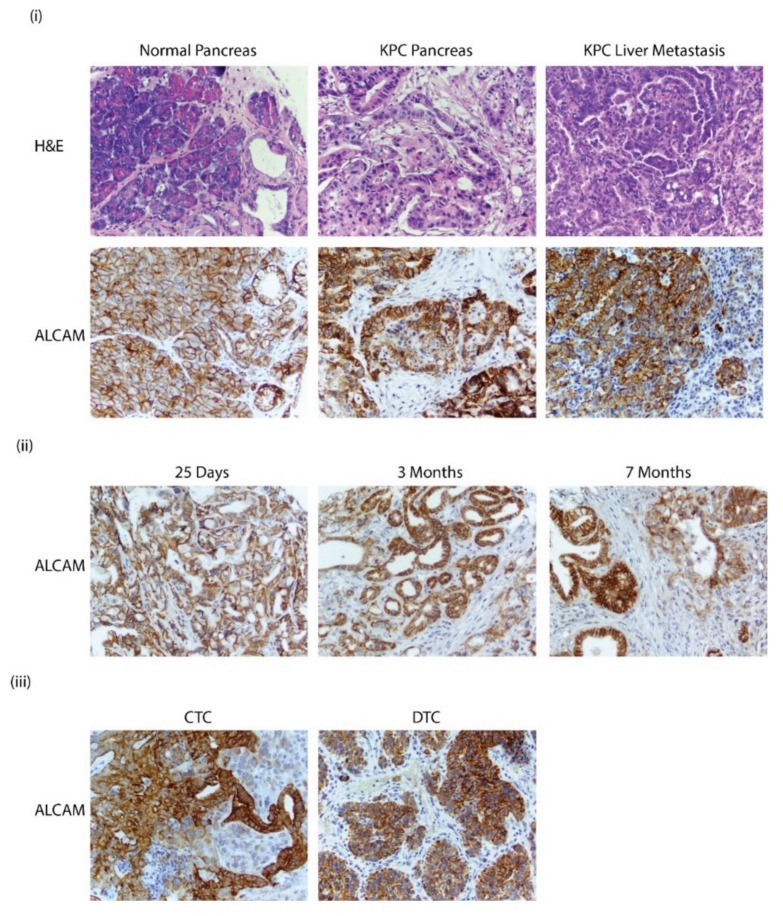
(**i**) H&E and Alcam staining of pancreas in C57BL6 and KPC GEMM mice with pancreatic cancer. Staining of liver metastasis in KPC GEMM mice (40× magnification); (**ii**) Alcam staining in KPC GEMM pancreas collected from mice that were 25 days, 2 months, and 7 months; (**iii**) Alcam staining in circulating tumor cell (CTC) and dissociated tumor cell (DTC) that were propagated in subcutaneous models (40× magnification).

**Figure 6 biomedicines-10-01983-f006:**
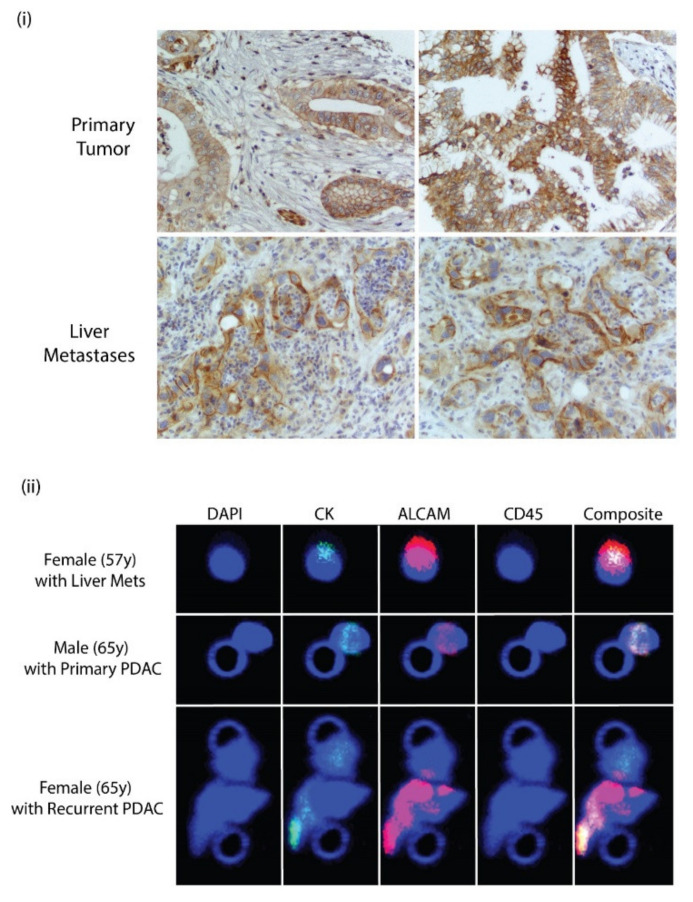
(**i**) ALCAM expression in human PDAC and liver metastases. (**ii**) Immunocytochemical staining of CTCs isolated from peripheral blood of patients with pancreatic cancer. CTCs were captured on Circulogix filters and stained for DAPI, cytokeratin, ALCAM, and CD45 along with a composite image of all stains. Patients were a 57-year-old female with liver mets, a 65-year-old-male with pancreatic ductal adenocarcinoma, and a 65-year-old female with a recurrent case of pancreatic ductal adenocarcinoma.

**Figure 7 biomedicines-10-01983-f007:**
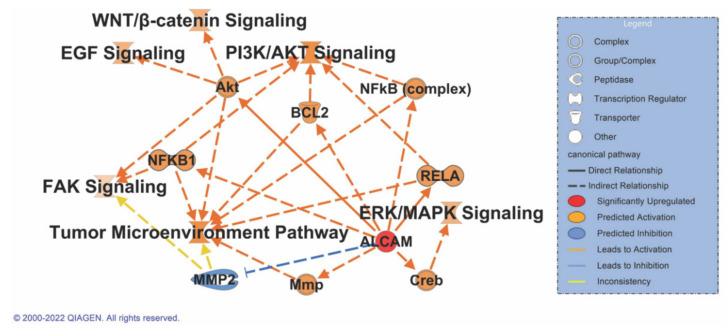
IPA path design with major signaling pathways specific to pancreatic cancer to be upregulated on activation of ALCAM. Solid lines indicate a direct association while dashed lines indicate indirect association with a signaling pathway. Orange denotes activation while blue indicates inhibition, and yellow indicates inconsistent relationship or data points to predict the connection between the nodes.

**Table 1 biomedicines-10-01983-t001:** Top 10 surface markers expressed on all CTC lines with more than 30% of one of the cell lines expressing the marker. The percentage positive fractions of each CTC line are indicated.

	Protein Marker	Gene Name	Description	CM61 (% + ve)	CF49 (% + ve)	HM59 (% + ve)
1	CD85H	LILRA2	Leukocyte Immunoglobulin-Like Receptor Subfamily A member 2	99.93	100	100
2	CD166	ALCAM	Activated Leukocyte Cell Adhesion Molecule	99.83	99.95	99.98
3	MICA/MICB	MICA/MICB	MHC Class I polypeptide-related sequence A/B	99.90	99.93	99.86
4	CD9	CD9	CD9 antigen	99.73	99.79	99.93
5	EPHA2	EPHA2	Ephrin Type A receptor 2	97.63	99.19	99.16
6	CD252	TNFSF4	Tumor Necrosis Factor Ligand Superfamily member 4	93.08	95.88	97.67
7	CD129	IL9R	Interleukin-9 Receptor	90.50	94.89	93.97
8	CD73	NT5E	5′ Nucleotidase	83.43	90.05	90.40
9	CD263	TNFRSF10C	Tumor Necrosis factor receptor superfamily member 10C	67.75	78.85	83.37
10	CD215	IL15RA	Interleukin 15 receptor subunit alpha	60.04	57.96	72.92

## Data Availability

All mass spectrometry and sequencing raw files along with analyses files have been deposited at BioStudies under the accession number S-BSST861.

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
