# Peer review of "ALCAM: A Novel Surface Marker on EpCAMlow Circulating Tumor Cells"

_biomedicines, 2022, doi:10.3390/biomedicines10081983_

Round 1
Reviewer 1 Report
Please see the detail comments in the attachment.

Author Response
"Please see the attachment."

Reviewer 2 Report
Well-structured paper.
I think that the scientific work behind the paper is well done,
The paper is paving the way for an alternative approach to pancreatic cancer testing, in particular in non-clear-cut cases.
Author Response
"Please see the attachment."

Reviewer 3 Report
The manuscript entitled:" ALCAM: A Novel Surface marker on EpCAMlow Circulating 2 Tumor Cells" focused on the identification of a nove lsurface biomarker in CTC from pancreatic cancer patients represents a timely relevant and technically correct manuscript suitable for publication on this journal after minor considerations:
- In the study design section, the authors highlight two simultaneous approach able to detect surface proteins in PC patients. Please, could the authors define the benefit of this approach for the study? Could they also show differences in terms of obtained results between these two methodologies?
- In the material and methods section, the authors report that 50 ml of blood was collected for each patients. In my opinion, this volume is not adequate to standardized this procedure on real world series due to low clinical status of patients. Could the authors explain why they started from this blood amount? Could they also discuss this point?
- In the text, the authors show robust results that confirm author's hypothesis. Accordingly, could the authors show the role of this novel detected biomarker in PC patients?
Author Response
"Please see the attachment."

Round 2
Reviewer 1 Report
The quality of the manuscript has been greatly improved. The conclusion could be strong if more samples from patient can be included in the analysis. The new figure 4 is good but the resolution is too low for some of the labels, especially for 4ii and 4iii, which should be improved. In general, the manuscript is in a good shape.
